# Culinary and Gastronomic Practices during the Periods of Restrictions on Movement Caused by the COVID-19 Pandemic in the Province of Alicante (Spain)

**DOI:** 10.3390/foods12152838

**Published:** 2023-07-26

**Authors:** Maria Tormo-Santamaria, Lluís Català-Oltra, Alexandre Pereto-Rovira, Ángeles Ruíz-García, Josep Bernabeu-Mestre

**Affiliations:** 1Carmencita Chair of Gastronomic Flavor Studies, University of Alicante, 03690 Sant Vicent del Raspeig, Spain; angeles.ruizgarcia@ua.es (Á.R.-G.); josep.bernabeu@ua.es (J.B.-M.); 2Balmis Research Group on the History of Science, Health Care and Food, University of Alicante, 03690 Sant Vicent del Raspeig, Spain; 3Department of Sociology II, University of Alicante, 03690 Sant Vicent del Raspeig, Spain; 4CRITERI: Critical Socioeconomics and Territory, University of Alicante, 03690 Sant Vicent del Raspeig, Spain; 5GASTERRA, Mediterranean Gastronomy Centre, University of Alicante, 03700 Dénia, Spain; alexandre@ua.es

**Keywords:** culinary skills, gastronomy, eating behaviour, food choice, COVID-19, confinement, Alicante province, Spain

## Abstract

Introduction: The third wave of COVID-19 had a large impact on the autonomous Region of Valencia, which gave rise to restrictions on movement and access to collective eating establishments. The objective of this study is to analyse the culinary and gastronomic behaviour exhibited by the population of the province of Alicante during the period of restrictions, in early 2021, in order to compare the results with an identical survey carried out during the first lockdown of 2020. Methods: observational and repeated cross-sectional study. Results: The frequency and time dedicated to cooking were similar, as was the tendency to cook as a family, although the percentage of meals ate alone increased and the presence of audiovisual devices during meals persisted. Recipes, cookbooks, websites and online courses became the principal sources of learning and the self-perception of improvements in culinary skills was greater. The cooking of traditional dishes of the Mediterranean diet predominated to the detriment of ready meals, but 41.6% of those surveyed preferred to improvise. The recipes most consulted were those for main courses. Conclusions. In spite of certain changes and setbacks, which in many cases led to a regression to the situation prior to the pandemic, many of the improvements made during the lockdown of 2020 persisted. Changes were made in culinary and gastronomic practices that can help to achieve a more conscious, healthy and sustainable diet but which require educational policies and actions to reinforce and consolidate them.

## 1. Introduction

In mid-January 2021, Spain was immersed in its third wave of COVID-19 [1,2]. The Region of Valencia, the autonomous region in which the province of Alicante is located, was one of the most affected areas. As stated in the preamble of the resolution of the Con-selleria de Sanidad Universal y Salud Pública (Regional Ministry of Universal Healthcare and Public Health) of 19 January 2021, the regional government proceeded to apply exceptional measures as a result of the worsening of the health crisis caused by COVID-19 [3]. As well as extending and expanding the measures agreed in the resolutions of 5 December 2020 and 5 January 2021, aimed at restricting the movement of the population [4,5], the preventive closure and precautionary suspension of activities in establishments and spaces related to the hotel and restaurant industry (bars, cafeterias and restaurants) were agreed. In this way, a period of restrictions began which lasted until March 2021 [6] and which, in many aspects, resembled the situation prevailing in 2020 due to the lockdown declared due to the first wave of COVID-19. Specifically, the consumption of food prepared in restaurant establishments fell again, and there was an increase in culinary activities practised in the home [7].

This occurred in a context in which, after the first lockdown and due to the restrictions on movement, consumption patterns had been modified. While certain alternative ways increased, such as home delivery or take away services, there was also a re-duction in the use of restaurants and other collective eating establishments [8].

The changes that occurred in dietary habits and the culinary and gastronomic practices during the first lockdown have been studied for different geographical areas, including the Region of Valencia [9,10,11,12,13,14,15,16,17,18]. In the case of culinary and gastronomic practices—we understand culinary and gastronomic practices as those that incorporate a series of know-how, techniques and specific knowledge together with the culture and history associated with the past of those who perform them [19]—different studies have highlighted the improvement in commensality practices, particularly among parents and children and the incorporation of the latter in culinary activities [20,21,22]. Other interesting findings were the positive association between the increase in the culinary activity, the improvement and acquisition of new skills and a better diet with an increase in the consumption of fruit and vegetables and a reduction in the consumption of fast food, in exchange for an increase in the preparation of traditional dishes of the Mediterranean diet. This pattern is considered as being exemplary of a healthy and sustainable diet [23]. The same was the case for eating empowerment, although this was often conditioned by socioeconomic status. Furthermore, different studies have indicated the need to reinforce these conducts and verify whether they have been maintained over time [24,25,26,27,28,29,30,31,32,33,34,35,36,37].

With respect to the changes observed during the lockdown of 2020, the following question was asked: “Which changes were maintained during the restrictions of 2021 and which returned to the situation of 2019?” In order to respond to this question, this study analyses the culinary and gastronomic behaviour displayed by the population of the province of Alicante during the period of restrictions prevailing in the Region of Valencia between January and March 2021 and compares these results with those provided by the survey conducted through social networks and local media between 28 April and 10 May 2020, in order to determine the changes that occurred in the culinary and gastronomic practices of the population of the province of Alicante during the first lockdown [17]. We have used an identical methodology to that of the first study. A survey was conducted with the principal objective of determining how many of the changes detected in the first survey have been maintained over time or the extent to which they have modified. The specific objectives of the survey were to determine the frequency and time dedicated to cooking, the motivations for cooking, the type of cuisine practised, with whom people cooked and the sources of information used, as well as the possible improvement of culinary knowledge and skills, in addition to finding out whether mealtimes were shared and with whom. Therefore, the aim is to determine the changes that have taken place and whether they are positive or not and how they can be consolidated or corrected.

## 2. Materials and Methods

An observational and repeated cross-sectional study was carried out. The diachronic orientation would not only be present in the repetition of the survey but also in some of the questions of the first test, which refer to the perception and experiences of the interviewees before and after the lockdown.

### 2.1. Universe and Sample

In the two information gathering exercises in April–May 2020 and January–March 2021, a non-probabilistic sampling of convenience was conducted, with a final adjustment to the real population through quotas of sex, age and level of education, using weighting coefficients. With this weighting adjustment and after having followed the same procedure of recruiting participants, the two samples are close to being equivalent. The universe comprises the population between 18 and 69 years of age residing in the province of Alicante, the fifth most populated of the 50 Spanish provinces, with almost two million inhabitants. It is also the fifth most densely populated province [38]. Therefore, this territory is highly characteristic of Spain’s urban areas, which are mostly concentrated on the coast, except for Madrid and a few other isolated cases. Given that the survey was conducted through the Internet, we decided to establish a maximum age limit of 69 because, according to data of the 2019 Survey on Equipment and Use of Information and Communication Technologies in Households of the Spanish National Statistics Institute [39], up to this age, the Internet is used by more than 80% of the subjects; however, from the age of 70 the proportion of users diminishes considerably and therefore the Internet users do not adequately represent all of their age cohorts.

### 2.2. Sequence of Fieldwork

In 2020, an ad hoc questionnaire was designed and divided into four parts. Through the different sections, socio-demographic information was gathered together with that referring to culinary and gastronomic practices prior and during the lockdown due to COVID-19 [17]. Similarly to the studies by Acosta-Banda et al. [40] or Napitupulu [41], in other fields, this initial questionnaire was qualitatively validated by the five members of the research team and other colleagues from the nutrition and gastronomy departments of the University of Alicante, who evaluated whether the indicators adequately covered the concepts described in the information objectives.

Section 1 described the characteristics of the study and provided the participants with information about the promoters of the research, in this case, the Cátedra Carmencita de Estudios del Sabor Gastronómico (Carmencita Chair of Gastronomic Flavor Studies), one of the institutional chairs of the University of Alicante. It indicated the objectives of the study and that the data provided would remain anonymous. Contact data was included for any additional information or consultations. It also informed of the maximum time required to complete the questionnaire, after previously testing its time range in a pre-test of 22 cases. After this informative briefing, Section 2 invited the participants to give their informed consent in order to access the following sections. Section 3 referred to socio-demographic items. Section 4 contained items with different response formats (dicho-tomous categorization, ordinal scale and numeric and alphanumeric open-ended questions) related to gastronomy during the period of restrictions, culinary and gastronomic practices, culinary skills and the participants’ assessment of the agro-food system. The items were selected by the members of the research team and expanded after being subjected to the prior scrutiny of experts including nutrition academics and gastronomy professionals who regularly collaborate with the Carmencita Chair of Gastronomic Flavor Studies. No previous or adapted questionnaire was used as there are none that include gastronomic questions related to an exceptional situation such as a temporary lockdown.

In 2021, the 2020 questionnaire was used as a base, creating a practically identical tool as that of the previous year so as not to lose comparability. Questions that did not provide significant information were eliminated and new ones were incorporated. This study focuses on the indicators that allow a direct comparison between the three periods (the two contemplated in 2020, before and after the lockdown, and that of 2021) and future research will work on the specific aspects of this survey.

The Google Forms application was used as an online support to gather information. In both tests, the participants were recruited via the Internet with diffusion on social media and local media channels. The first data gathering exercise was conducted between 28 April 2020 and 10 May 2020, the latter being the starting date of Phase 1 of the so-called “de-escalation” of restrictions in certain territories of the province of Alicante and in Spain as a whole. A total of 1045 people responded to the survey, but 5 cases were eliminated as they did not complete at least 80% of the questionnaire, 21 due to duplicated answers, 2 due to inconsistent responses, 42 cases of people over the age of 69, and finally, 296 cases of people residing in other provinces. Therefore, the final sample included 679 cases.

In the second test, the information was gathered between the 15 March 2021 and the 17 April 2021. A total of 523 people responded. However, 5 cases were eliminated as they did not complete at least 80% of the questionnaire, 10 due to duplicated answers, 2 due to inconsistent answers, 17 of people over the age of 69 and, finally, 84 cases of people residing in other provinces. Therefore, the final sample included 385 cases. This was noticeably smaller than the sample of 2020, when the participants responded to the questionnaire in what was an idle period for many inhabitants; however, it is sufficient for a study of these characteristics. The figure is equivalent to that which usually corresponds to probabilistic samples for a confidence interval of 95.5% and a margin of error of ±5%.

It should be noted that, although the first data collection exercise began at the end of April, it refers to the lockdown period, which began in mid-March and included the Easter holidays, when it is traditional to make baked or fried cakes and sweets in this territory. Meanwhile, the second survey was conducted in a period beginning in mid-January when the population was exercising more control over what they were eating as they had just finished celebrating the Christmas holiday with its dietary excesses. This should be taken into account when interpreting the data.

### 2.3. Deontological Aspects

The research protocol was carried out in accordance with the Helsinki Declaration for research involving human subjects by the World Medical Association, strictly respecting the confidentiality of the information in accordance with Organic Law 15/1999 of 13 December on the Protection of Personal Data in all of the gathering and treatment processes of the information obtained and Organic Law 3/2018 of 5 December on the Protection of Personal Data and the Guarantee of Digital Rights. The study was approved by the Ethics Committee of the University of Alicante (Dossier UA-2021-03-15, 30 March 2021).

### 2.4. Considerations of the Analysis

The data were analysed using the IBM SPSS Statistics 25 package. It consisted of comparing the practices prior to the emergence of COVID-19 and those that prevailed during the two periods of restrictions on movement, one stricter (2020) than the other (2021). In relation to the first data gathering exercise, although it was a strictly observational study, to some extent it also constituted a pre-test/post-test pre-experiment of a group [42], although with the difference that in this case, the data gathering was not conducted at different times, but only once but of information referring to two different periods, before and during the lockdown, which are the two periods compared. Logically, the (pre)experimental situation is the lockdown, which has aroused enormous academic interest due to its exceptional nature and because it constituted a situation which, in one way or another, completely or partially affected the vast majority of the population. The situation prior to the pandemic acts as a reference of normality with which to compare the exceptionality through the indirect observation of a recollection of a recent time in which a habit was formed. Evidently, there is a bias in perception (the actions of a subject are not the same as the recollection that he or she has of them) and it is also logical that distance in time can distort the memory. However, this time distance is short and the aggregate of the perceptions of the different subjects converge into an average that is very close to reality, as documented by Rechenchosky et al. [43] when referring to the collection of height and weight data in a survey both verbally and through direct measurements with precision instruments: the difference between the values of one method and the other is minimal. In this respect, we understand that this recollection of the previous normality is a valid reference for the objectives of the research.

In order to statistically examine the differences between one period and another, it has been assumed that the samples are related: the same questions are asked for both periods with the same indicators and the same people, therefore, each pair of indicators has a high probability of being associated. Accordingly, in order to identify whether the differences between one period and another are statistically significant, we should consider the assumption of paired or related samples. Therefore, based on this and the confirmation of non-normal distributions (as indicated by the Kolmogorov–Smirnov Z test, with significance levels below 0.00001 in practically all of the key variables), Wilcoxon’s rank-sum test has been used for ordinal variables and the McNemar test for nominal variables. In both cases, we consider differences to be significant when the significance levels were lower than 0.05, in the same way as Coll-Planas et al. [44].

Comparing these two periods by contrasting the first set of information gathered with the second set of information implies the use of independent samples. Considering non-normal distributions (as also indicated by the Kolmogorov–Smirnov Z test, with significance levels below 0.00001 in practically all of the key variables), we used the non-parametric Mann–Whitney U test in the case of ordinal variables and the chi-squared test in nominal variables. In both cases, the differences were considered to be significant when the significance levels were lower than 0.05, similarly to other researchers [45,46]. It is considered that non-significant differences indicate the similarity between the groups or situations (before COVID-19) during the 2020 lockdown and during the period of restrictions at the beginning of 2021).

## 3. Results

This section may be divided by subheadings. It should provide a concise and precise description of the experimental results, their interpretation, as well as the experimental conclusions that can be drawn.

### 3.1. Comparison of Culinary and Gastronomic Habits before the Emergence of COVID-19, during the Lockdown at the Beginning of 2020 and the Period of Restrictions at the Beginning of 2021

During the lockdown, both the frequency with which the participants cooked and the time that they spent cooking increased (Table 1, ^a^ and ^b^. In fact, a parameter such as the median, which is not sensitive to variations in short ordinal scales, increases in the first two variables of Table 1, from 2 (“a few days a week”) to 3 (“every day”) in frequency, and from 1 (“less than 1 h/day”) to 2 (“between 1 and 3 h/day”) in time spent cooking. The availability of time (82%) and less rigid timetables (44.4%) enabled this considerable variation in accordance with the principal motivations to increase culinary practices expressed by the participants [17].

The differences between the habits prior to the state of emergency and the behaviors adopted during the 2020 lockdown are statistically significant, both in the overall frequency of cooking during the lockdown and in the time spent cooking, as we can observe in the practically residual levels of statistically significance which, therefore, leave little room for doubt.

Meanwhile, and still with points 1 and 2 of Table 1, if we compare these results with those of the 2021 survey, referring to the period of restrictions in the third wave of COVID-19 in Spain, we can verify that the response pattern is more similar to that of the strict lockdown of 2020 than a normal period. This is confirmed by the levels of significance which are very low in the case of the comparison between the situation of 2019 and that of 2021, but relatively high in the comparison between 2020 and 2021, which implies similarity between the two distributions.

Cooking “every day” which was not a habit of the majority before the lockdown (45.7%), became widespread behavior during the lockdown (almost 70%), but also during the period of restrictions in 2021 (72.2%). The majority (52.4%) cooked for less than one hour before the lockdown, but with the state of emergency and the movement restriction measures that it brought about, almost two thirds of the population of the province of Alicante interviewed increased the time spent on cooking to between 1 and 3 h per day and this was maintained in 2021 during the third wave of COVID-19.

In the rest of the indicators of Table 1 that reveal the details of cooking (sources of learning, type of cuisine, type of dishes and socialization), obtained through the McNemar test for all of the categories (taken as dichotomous variables), the differences between “before” and “during the 2020 lockdown” are statistically significant (bilateral exact significance <0.05) in all cases except “Online cookery course,” “Improvised,” and “Other type of cuisine,” “Appetizers,” and “Mother,” “Father” and “Others.” Therefore, in general, significant changes were brought about by the lockdown situation.

With respect to the relationship between the survey of 2021 and the two periods of the 2020 survey, in sources of learning (Table 1 ^c^) the differences are significant with these two periods, but the trend observed in the 2020 lockdown is reinforced, so the family environment has given way to cookbooks, recipes, websites, etc., which clearly constitute the majority source, and online courses are overtaking face-to-face courses.

### 3.2. Type of Cuisine

The differences in the type of cuisine prepared before and after the 2020 lockdown are statistically significant (Table 1 point 4). In both periods “traditional (Mediterranean)” cuisine clearly dominates but experienced a slight upward shift during the lockdown (around six percentage points) in detriment to ready meals, whose percentage reduced by more than half. In the period of restrictions in 2021, the ready meals maintained the lowest levels, but after the learning period of the 2020 confinement, in 2021 there were a considerable number of interviewees (41.6%) who had shifted to improvised cuisine.

The results of 2020 also show that not only had people cooked more, but they had cooked a wider variety of cuisines (Table 1 point 5). The most substantial changes occurred in “desserts” (from 20.8% before the lockdown to 35.5% during it) and “savoury baking” (from 7.7% to 14.5%). In 2021, on the other hand, the respondents of the study clearly returned to principally focusing on main courses.

### 3.3. Cooking and Sociability

As we can see in the results (Table 1 point 6), during the 2020 lockdown, cooking in the company of others increased: cooking “alone” decreased by more than 15 points during this period of maximum restrictions, while all of the group cooking categories increased, particularly “partner” and “children” (around 11 percentage points in both cases). This trend was maintained in the period of restrictions of 2021 for these two categories and, therefore, the indicator “who people cook with” is similar for the two periods of restrictions, (stricter in 2020 and laxer in 2021), unlike the normal period prior to the confinement, which is confirmed by the levels of significance of the chi-squared tests.

Those percentages for “Before COVID (2019)” or “During the COVID lockdown (2020)” that are closer to those obtained for “During the period of COVID restrictions (2021)” are highlighted in blue.

As we can observe in the results reflected in Table 2, the meal least altered by the pandemic (both in 2020 and 2021) is dinner (as revealed by the significances of the McNemar and chi-squared tests). The changes occur in the other meals, particularly during the 2020 lockdown, with an increase in the practice of eating in company in all of them due to the forced co-habitation as a result of the restrictive situation. Although the practices during the period of restrictions of 2021 were far removed from both the situation prior to the pandemic and the lockdown, some of the features of normality of 2019 were recovered.

With respect to carrying out other activities, the increases in eating in company are always higher when television is included, and the decreases in eating “alone” are lower when other things are completed at the same time (particularly in the 2020 lockdown, but also in the period of restrictions of 2021). Therefore, during the pandemic, people ate to a greater extent in company but also with a greater presence of audiovisual devices.

The mid-morning snack and/or afternoon snack were those meals that were usually skipped by a greater proportion of the population both during the lockdown and in the situation of normality. The change brought about by the 2020 lockdown is that the percentage of interviewees who skipped the mid-morning snack increased but that of those who skipped the afternoon snack decreased. In 2021, the respondents of the study showed a slight tendency to skip fewer meals, but their conduct with respect to the mid-morning snack was more similar to that of the pre-COVID-19 period while that of the afternoon snack was more similar to the 2020 lockdown.

### 3.4. Self-Perception of Cooking Skills

Although in both periods there is a relatively higher percentage of participants who did not perceive any improvement (18.3% and 15.1%; see Figure 1), the large majority, around two-thirds, declare an average or high level of improvement in their culinary skills during the confinement (from 5 to 10) and a high percentage give scores of between 7 and 10, which implies a significant level of improvement. However, these high or excellent self-assessments were somewhat higher in 2021 (45.2% as opposed to 32.1% in 2020) which gives rise to statistically significant differences in the mean, as we can see in Table 3.

## 4. Discussion

The results reveal that during the third wave of COVID-19, the population of the province of Alicante increased the frequency and time spent cooking in a similar way to the strict lockdown of 2020. Therefore, this suggests that certain new conducts that emerged during the 2020 lockdown [17] were maintained, even when the restrictions on movement were less rigorous. These data reinforce the importance that nutrition has acquired during the pandemic and its protagonism in everyday life [12].

Another of the results worth highlighting is the relationship between the survey of 2021 and the two periods of the 2020 survey with respect to learning sources. There has been a reinforcement of the trend emerging in the 2020 lockdown whereby the family environment gave way to cookbooks, recipes, websites, etc., which now clearly constitute the majority source. The online courses are overtaking face-to-face courses, both due to the health safety requirements which have led to a decline in social contacts and an increase in digitalization [47,48].

In the period of restrictions of 2021, as well as a clear predominance of traditional Mediterranean cuisine to the detriment of ready meals [28], there was a considerably high percentage of participants (41.6%) who began to improvise. This is a characteristic of cooks with more experience [25,30]. In this sense, we can affirm that the lockdown period of 2020 served as a learning phase with respect to the type of cuisine, which gave rise to creativity in the framework of greater eating empowerment [33,49]. This trend may contribute to the achievement of a healthier diet [26,49].

During the 2020 lockdown, within the variety of dishes prepared, sweet and savory baking were particularly noteworthy. However, in 2021, the recipes most consulted were those of main courses. This result may be influenced by the fact the questionnaire was administered in different seasons of the year. The first survey coincided with a season when baking products are most frequently consumed [17]. However, the second survey was conducted after Christmas, when the population is more prone to watching what they eat in order to control their weight. This change in culinary habits can also be explained by the recovery of routine and timetables which helped to avoid certain inappropriate eating habits developed during the 2020 lockdown period [9,10,11,12,13,14,15,16,20,26,29,50,51,52,53].

Although the availability of time was the main reason for the participants to cook more, it was also important to be able to cook as a family. This circumstance was maintained in the period of restrictions of 2021, which indicates an interest to involve the whole family, in particular the children, in preparing meals [20,21,54,55,56]. Therefore, the interest shown by the interviewees in teaching and learning to cook could be the object of future educational actions that consolidate these practices and can help to improve the eating habits of the population [57,58]. These educational actions should include food and nutrition, as well as culinary and gastronomic content [59], with the participation of dietitians together with that of gastronomes, farmers and pedagogues. Furthermore, they would require the joint efforts of the Ministries of Education and Professional Training, the Ministry of Health, the Ministry of Agriculture, Fishing and Food, among other institutions. For this specific case, it should be noted that some of the changes made during the period of restrictions in 2021 were unstable. While some seem to have been maintained and have become spontaneously stable, others have gradually regressed back to the normal situation of 2019. Therefore, the educational changes that are carried out should take into account this duality. In this way, for those changes that have been spontaneously maintained, the educational action would not involve too much dedication, being limited to reminders and a reinforcement of the appropriate conducts. This would allow the educational efforts to be concentrated on the recovery and consolidation of those behaviours that could improve the empowerment of the population through culinary and dietary habits. Activities should be developed that are based on the promotion of the cooking of traditional dishes, typical of the Mediterranean diet, using the territory as a tool to indicate proximity or zero km products, create and use school vegetable gardens, promote inter-generational dialogue or involve the whole family [55,59,60,61].

With respect to convivial eating, in the period of 2021, we can observe an increase in the percentage of people who ate a mid-morning snack and also a step backwards with respect to the advances detected during the 2020 lockdown. The “eating alone” category increased in all meals. This circumstance, similarly to the higher frequency of mid-morning snacks, can be explained by a greater flexibility of the norms regulating movement. We should remember that during this period or restrictions, a greater percentage of the population were working than during the 2020 lockdown [3,4,5,6]. Furthermore, the improvements in conviviality which have been maintained or did not diminish excessively in the period of 2021 with respect to the 2020 lockdown are limited to those situations where people eat in company but with the presence of audiovisual devices. The need to contact relatives outside of the home (for example, through social networks) and to access information about the pandemic situation were surely at the origin of the increase in the use of these technologies and also the aggravation of addictive conducts with respect to the use of these devices [62,63,64]. We must not forget that commensality, that is, eating and drinking together at the same table, represents one of the core elements of sociability as it enables the sharing of food as a social ritual and the creation of a symbolic space where a cultural identity is shared and transmitted. When addressing the phenomenon of diet, just as important as determining what we eat and what we should eat is with who and where we eat, among other factors [65].

Finally, the people surveyed in the period of restrictions of 2021 declared a self-perception of improved culinary skills greater than during the 2020 lockdown, maybe because during this second period they were able to gain knowledge about the benefits acquired in the previous period. This constitutes a potential improvement that, although subjective, may positively contribute to the achievement of a healthier and more conscious diet [25,31,33,49], exercising the caution required when handling information drawn from self-assessment.

In any event and in general, this second period, with less restrictions than the first, is confirmed by many indicators as being a mixed situation, half-way between normality and the radical exceptionality of the largest lockdown in the history of humankind. Future research should address this issue and others in order to analyse the changes in dietary and culinary behaviour of the population. In order to carry out this research, it should be taken into account that the period when the data for this study and the previous study were collected was characterised by the public measures in force to combat the COVID-19 pandemic. Therefore, future data collection should be performed at logical moments that will enable the changes in culinary and gastronomic practices of the survey participants to be analysed and understood when not influenced by the pandemic. The time of year when the data were collected for the two studies should be the same in future studies in order to minimise the seasonal effect. We hope that this future research will confirm some of the trends that have been hypothetically indicated, as this would mean that improvements have been made in the habits of the citizens.

One limitation of this study is that the same sample size of the first survey could not be obtained for the second, maybe because during the 2020 lockdown, in general, people had more time and were more willing to answer this type of questionnaire. In any case, as we have already mentioned, it is a sufficiently large number. With regard to the strengths of the research, we can highlight the incorporation of culinary and gastronomic practices as a principal variable of analysis, which has undoubtedly enabled us to reveal the influence of culinary activities and gastronomic habits on the dietary changes that have occurred during the periods of restriction on movement brought about by COVID-19. In many cases the novelties are related to an improvement in the attitudes of the participants towards nutrition. This circumstance can be taken advantage of for the implementation of future educational actions in food and nutrition education but also in culinary and gastronomic education, in order to empower the population so as to consolidate and reinforce the positive changes in behaviour that took place in the periods of restrictions in 2020 and 2021.

## 5. Conclusions

The culinary and gastronomic practices that characterised the eating behaviour of the surveyed population of the province of Alicante during the period of restrictions on movement due to the third wave of COVID-19 are in line with the most important changes detected in the study of the first lockdown, which compared conducts in relation to the pre-COVID-19 situation. Although changes were made to the type of dishes prepared or setbacks occurred, such as an increase in meals eaten alone or accompanied by audiovisual devices and greater improvisation in cooking, the persistence of many of the improvements in culinary and gastronomic practices detected during the lockdown of 2020 represents an opportunity to advance in the objective of a more conscious, healthy and sustainable diet. To do this, it would be necessary to implement educational policies and actions that consolidate the attitudes and habits related to the importance given to diet, stimulate interest in improving culinary skills, cooking more often and doing so in company or preparing traditional dishes typical of the Mediterranean diet rather than ready meals.

## Figures and Tables

**Figure 1 foods-12-02838-f001:**
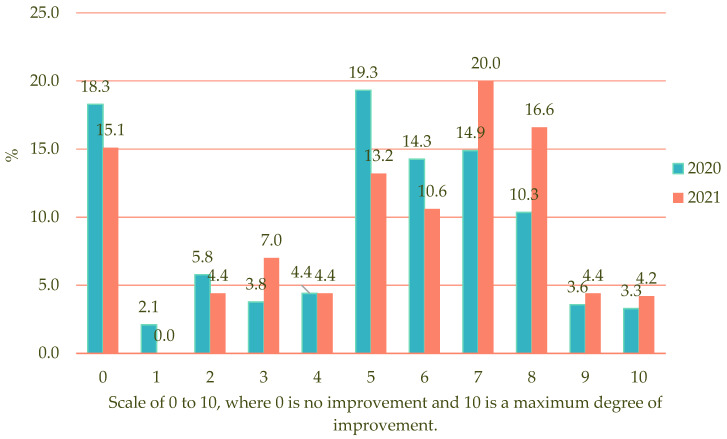
Self-perception of improvement in culinary skills in the COVID-19 period of restrictions.

**Table 1 foods-12-02838-t001:** Culinary and gastronomic practices before the emergence of COVID-19, during the lockdown at the beginning of 2020 and the period of restrictions at the beginning of 2021 (%).

	Before COVID-19 (2019)	During the COVID-19 Lockdown (2020)	During the Period of COVID-19 Restrictions (2021)
1. THE FREQUENCY WITH WHICH PEOPLE COOKED ^a^ *^,^ **			
Never	5.7	3.9	4.9
At weekends	12.2	3.8	5.2
A few days a week	36.3	23.3	17.7
Every day	45.7	69.1	72.2
Total	100.0	100.0	100.0
Median	2	3	3
2. TIME SPENT COOKING ^b^ *^,^ **			
Did not cook	5.8	3.5	4.9
Less than one hour/day	52.4	20.9	21.6
Between 1 and 3 h/day	35.5	63.5	66.4
More than 3 h/day	6.3	12.2	7.0
Total	100.0	100.0	100.0
Median	1	2	2
3. SOURCES USED TO LEARN HOW TO COOK ^c^ *^,^ **^,^ ***			
Family environment	80.6	53.7	35.8
Cookbooks, recipes, blogs, TV, social networks...	36.8	58.1	77.7
Face-to-face cookery course	8.4	6.1	2.5
Online cookery course	2.9	4.2	9.9
4. TYPE OF CUISINE ^c^ *^,^ ***			
Traditional (Mediterranean)	79.7	85.6	68.3
Improvised	29.0	28.9	41.6
Ready meals	10.1	4.6	4.2
Other	3.8	4.9	7.0
5. TYPE OF DISHES ^c^ *^,^ **^,^ ***			
Main courses	87.2	90.4	91.2
Appetizers	32.1	34.2	24.7
Desserts	20.8	35.5	22.6
Savoury baking	7.7	14.5	7.3
6. WHO PEOPLE COOK WITH ^c^ *^,^ **^,^ ***			
Alone	74.6	59.3	59.3
With partner	26.5	37.5	40.5
Children	7.7	18.6	15.4
Mother	8.9	9.2	5.5
Father	3.4	3.2	0.5
Other	1.9	2.1	1.8

* Significant difference between before and during the lockdown. ** Significant difference between before the lockdown and during the restriction period in 2021. *** Significant difference between during the lockdown 2020 and during the restriction period in 2021. ^a^ Scale: 0 = never, 1 = weekends, 2 = some days of the week, 3 = daily. ^b^ Scale: 0 = never cooked, 1 = less than 1 h/day, 2 = between 1 and 3 h/day, 3 = more than 3 h/day). ^c^ The sum of the percentages does not equal 100, as the question is a multi-response question.

**Table 2 foods-12-02838-t002:** Who people shared the different meals with before the emergence of COVID-19, during the lockdown at the beginning of 2020 and the period of restrictions at the beginning of 2021 (% valid for each meal).

	Before COVID-19 (2019)	During the COVID-19 Lockdown (2020)	During the Period of COVID-19 Restrictions (2021)
Breakfast			
Alone	61.0	44.4	49.6
Alone, doing other things	16.8	17.9	23.1
In company + TV	5.5	13.9	9.9
In company without TV	11.2	19.3	14.5
Did not eat this meal	5.5	4.5	2.9
Total	100.0	100.0	100.0
Mid-morning snack			
Alone	36.5	27.7	35.2
Alone, doing other things	14.6	15.8	22.9
In company + TV	9.1	13.3	9.4
In company without TV	20.6	16.8	9.9
Did not eat this meal	19.2	26.3	22.7
Total	100.0	100.0	100.0
Lunch			
Alone	15.7	8.1	14.0
Alone, doing other things	18.8	15.0	15.1
In company + TV	40.3	49.7	48.3
In company without TV	23.2	27.0	22.3
Did not eat this meal	1.9	0.2	0.3
Total	100.0	100.0	100.0
Afternoon snack			
Alone	35.2	23.7	37.4
Alone, doing other things	14.8	16.5	18.7
In company + TV	6.6	17.4	10.6
In company without TV	11.2	16.3	7.0
Did not eat this meal	32.3	26.1	26.2
Total	100.0	100.0	100.0
Dinner			
Alone	11.5	9.9	13.0
Alone, doing other things	15.6	14.2	16.6
In company + TV	51.0	53.7	51.2
In company without TV	20.4	20.6	18.7
Did not eat this meal	1.3	1.7	0.5
Total	100.0	100.0	100.0

Note: Exact bilateral significances in the McNemar test to compare the related samples (before-during the lockdown): Breakfast = 0.0000001, Mid-morning snack = 0.0000001, Lunch = 0.0000001, Afternoon snack = 0.0000001, Dinner = 0.188. Significances of the chi-squared test for independent samples lower than 0.03 in “Breakfast”, “Mid-morning snack”, “Lunch” and “Afternoon snack” and higher than 0.2 in “Dinner”, both in the relationship before COVID-19-Restrictions ‘21 and in the relationship lockdown ‘20-restrictions ‘21.

**Table 3 foods-12-02838-t003:** Self-perception of improvement in culinary skills during the periods of restrictions associated to the COVID-19 pandemic.

	Year of Survey	N	Mean	Standard Deviation	Standard Error of the Mean
Level of improvement in culinary skills	2020 lockdown	679	4.77	2.940	0.113
	2021 restrictions	385	5.32	2.929	0.149

Levene’s Test for the equality of variances: F = 0.002; signif. = 0.964. T Test for the equality of means in independent samples assuming equal variances: t = −2.975; signif. = 0.003.

## Data Availability

The datasets generated for this study are available on request to the corresponding author.

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
