# Peer review of "Culinary and Gastronomic Practices during the Periods of Restrictions on Movement Caused by the COVID-19 Pandemic in the Province of Alicante (Spain)"

_foods, 2023, doi:10.3390/foods12152838_

Round 1

Reviewer 1 Report

This study investigates the culinary and gastronomic practices during the periods of COVID-19 pandemic. Some enhancements can furtherly stress the importance of the results. The paper seems very weak, especially in terms of methodology. Suggestions are reported below.

1.       The term gastronomic practices exist in literature?

2.       What were the questions related to gastronomic practices?

3.       Questionnaire was validated?

4.       Is there culinary and gastronomic standard criteria?

5.       Attach the questionnaire at the end of article?

6.       The authors also did not provide sufficient evidence on literature review to support the hypotheses.

Reviewer 2 Report

This article is clearly written and presents the results of an informative but rather descriptive study on an issue of great interest. It concerns the changes made in consumption patterns during the COVID19 pandemic by comparing the pre-pandemic situation with the post-lockdown situation of 2020 and the post-restrictions situation of 2021 in the Alicante area in Spain.

However, no scientific hypotheses on the on-going situation is emitted. And the research status of this work would be supported by a focus on learning processes and it is not really the case. The missing hypothesis is replaced by a scenario (line 62), that of maintaining observable practices between 2020 and 2021 in the context studied. As a matter of facts, there is little mention of other situations studied and it is not known how Alicante is positioned in relation to other situations in Spain and elsewhere.

The paper therefore presents relevant references that will be useful to perceive the ruptures and continuities over the long term attributable to COVID19 and its containment measures. But it does not seem to produce enforceable knowledge or enrich a corpus on the eating habits of the populations concerned.

The data collection work is well done, in accordance with the usual precautions in such work. I have not noticed any particular problem in statistical processing but I am not an expert in this field and other reviewers will probably be able to clarify this point.

In line 57, fruit and vegetable consumption is mentioned in relation to healthier diets, and everybody can agree with this statement. However, fruits and vegetables are never mentioned again in the survey or in the results obtained. It would have been useful to know to what extent respondents had taken this need for fresh fruit and vegetables into account in their diets. The word “traditional” seems too fuzzy for providing any certainty in this matter. As a result, talking about “healthier diets” as it is done several times in the text remains a poorly documented shortcut. On line 361, the reader can actually wonder how the authors claim they observe so-called “healthier diets”.

The context of the study is poorly presented. We can notice on line 180 some information about Alicante which is qualified as one of the most populated regions of Spain. But this passage is too short and it is curiously positioned at the end of the material and methods in the "data analysis" part. More information is needed and higher up in the paper to better understand the contributions of this study. Is Alicante area given as representative of the highly populated areas of Spain? Are there contrasts with more rural areas?

The seasons of the different surveys play an important role in the collection of certain data. They are not presented in material and method while their differences between 2020 and 2021 are mentioned in lines 364 and following under discussion as one of the limitations of the study. They should be presented in part 2 for avoiding the reader only discover this point in the discussion.

The number of respondents is almost halved (385 vs. 679) between the two periods of the 2020 lockdown and the 2021 restrictions. In line 136, the authors describe this decrease as "slight" and in line 137, they pretend that it is not a problem. Yet, in line 396 of the discussion, they acknowledge that this is a difficulty for the reliability of the results. What reason makes it possible to say that 385 respondents is a "sufficient" number? And what is the loss of precision caused by this decrease in the size of the sample?

In line 249 of the results, it is said that the responses obtained are more similar to those of 2020 than to the situation before COVID19 of 2019. The authors deduce that a large part of the changes induced by the lockdown period are still observable, so the scenario of maintaining the new behavior is confirmed. However, could we not deduce, conversely, that after the forced and sudden changes of 2020, the situation is slowly returning to normal? Indeed, their deduction is more assertive than demonstrated. A contrary hypothesis of resilience of pre-pandemic habits could be made, with relative inertia in this return to normal. This is what we observe with dinner, which seems very little affected by successive situations and testifies to this strength of normality. Moreover, in line 369, we can notice that some changes in 2020 seem “inappropriate” and that the beginning of return to normal may correct these changes.

The authors should tell us why they believe their results express an even incomplete continuation of the 2020 changes rather than an incomplete return to 2019. And, in order to be sure, will it not be necessary to schedule a new study in 2023 (ie now) and examine where the movements of behavior are? Unfortunately, no prospect of future work is proposed.

In line 281 of the results, the changes are well referred to learning processes that should be the heart of the analyses and the object of knowledge of such a study. The data collected does not seem to give sufficient space to these learning processes in favor of notions such as "cooking alone" or "with whom". It is never easy to study the on-going changes, but the authors could seek to account for what matters in the changes made. For example, why are they interested in whether respondents cook alone or not? And what is the point of knowing if the meal is taken with a television set or not? Such data are unfortunately not really discussed afterwards.

"Cooking skills" are a particularly interesting point: the declarations line 330 suggest that they are maintained or progressing strongly. But it would be necessary to distinguish these two situations because, to cope with the sudden changes in situation (the lockdown of 2020), the respondents of the previous study had to deal with these situations in a very short time. In 2021, the population faced fewer constraints and with the gains of the previous period. However, learning about cooking is not so mobile: you can learn recipes, try them, abandon them, take them back as you need them again. In addition, the respondents answer a self-assessment that remains unreliable. These data should be called "cooking skills representations" or "self-assessment of cooking skills" to make it clear that this is not really a study of skills.

The “educational actions” mentioned in line 376 and which are found at the end of the conclusion (in line 405) are the main options identified by the authors. This very important point deserves to be explained and clarified: what are the courses of action that the authors identify? How are they likely to consolidate positive changes and correct inappropriate changes that are observed? And are they necessary to avoid a return to the pre-pandemic situation and reinforce the positive nature of the learning required by management measures of COVID19? Perhaps, putting this research object as the top of the authors’ preoccupations may reorient the paper toward a more scientific purpose and some consistent hypothesis.

Corrections:

Line 392: put 2021 instead of 2020 because we are talking about restrictions and not lockdown.

Reviewer 3 Report

The manuscript tackles a very interesting question, if diets and cooking behavior has changed due to COVID 19, and if these changes are having an influence to post COVID 19 times. Unfortunately, the Authors use a period (2021) where we had still a lot of restrictions and the time lag between the first measures (lockdown etc.) was short. Meanwhile we are mid 2023, so why not analyzing and comparing actual behavior? This issue is not solvable with the existing data. But besides that, there are other important shortcomings that have to be addressed (and that could be solved).

There is no information available about the structure of the sample. This is an important shortcoming, as the results of two samples are compared. Even though the Authors used a convenience sample, it is a usual approach to deliver the structure in view of socio-demographics. And I highly recommend to compare the structures of all samples in comparison to the overall structure of the population of Alicante.

The empirical design of the study is not really clear to me. The Authors refer to the original study [17] which was obviously done by the same research team. I assume, diets and cooking behavior was researched also before COVID 19, but the Authors are not really including this body of knowledge into their manuscript. Therefore, I am not completely convinced that the ad-hoc questions used in both studies are the best way to get answers to this interesting research question.

In particular, we have to consider that social desirability is a matter of fact when it comes to diets, cooking and eating behavior and self assessed behavior might deviate from observed behavior (which is much more difficult to be assessed, of course). Therefore, I suggest to at least to integrate this problem in the discussion section. I would be beneficial for readers to get access to the original questionnaire of the study.

Concerning limitations: the smaller number of respondents is the least restriction of the study IMO. It is a convenience sampling approach that is used to gather the required information. Therefore, the results are rather not transferable to the overall population of Alicante; this important limitation should be discussed in the limitations section. And I already mentioned the restrictions due to the ad-hoc study design.

Due to these limitations, the results should be addressed to "respondents of the study" and not "residents of ...". The results are, as I said, probably not transferable (a comparison of the structure mentioned above would be helpful here).

The way the Authors describe significant differences in the tables is unusual and hard to get. It would be better to integrate sig. diff., e.g., by use of *,**,*** for p < 0.5, 0.01, 0.001 in the tables itself. The description of the differences in the footnotes is very hard to follow. In my opinion, it should not be left to the readers to realize which answers are significantly different and which are not. The way of visualizing these differences (a central part of the study) has to be improved.

Referring to Table 1 a) (L235) etc. is not consistent with Table 1; there is no a, b, etc. The meaning of blue cells in Table 2 is not self explaining; and there is no reference to the blue cells in the text.

In Figure 1, only the sample of 2020 is visualized, why not in comparison to the data from 2021 (as it was done in Fig. 2)? Wasn't that part of the questionnaire? And are these differences (if available) significant?

Some of the arguments in the discussion are pointing towards invalid comparability of the results. If it comes to sweets, e.g., "This result may be influenced by the fact the questionnaire was administered in different seasons of the year." is rather true in my opinion. It makes a huge difference if cooking refers to baking, if respondents are asked before Christmas (THE season for baking) or after (the LEAST season for baking). Therefore, the comparison is meaningless. So, the interpretation in L284 is statistically true but doesn't tell us anything about a change in culinary habits and cooking behavior (due to COVID). The Authors should really critically re-analyze their results (not only these one) and thoroughly rethink if the results might be due to a real change in behavior or due to other limitations.

Some minor spelling errors, e.g. double spaces, "Va-lencia" etc. in abstract, "et al" without dot, etc. If you use the phrase "on the other hand", "on the one hand" should be part of the text before as well (e.g., L180). 2921 instead of 2021 (L307).

In general, the English language seems to be good, with the exception of some minor errors mentioned above.

Round 2

Reviewer 1 Report

  • Accept in present form

Author Response

Thank you for your contributions and for accepting the paper for publication. We have attached a new version of the article from the second round of revisions.

Reviewer 2 Report

The manuscript that is being submitted for the second time incorporates much of the remarks and suggestions from my previous revision. From my point of view, it is therefore more in line with my idea of a scientific article.

Without going back over the changes made along these lines, I will focus only on the problems that remain.

On the summary:

In L.17, I suggest adding after "the period of restriction" the following clarification: "on early 2021". This makes it possible to locate the new data that are studied in the article.

In L.25-26, I do not agree with the conclusions. It should be noted here that the situation in 2021 shows a combination of changes in 2020 and a return to "normal" in 2019. And secondly, to indicate that the consolidation of novelties, when they go in the direction of a healthier diet, will require educational actions.

On the text:

In L.67, I still do not understand why the authors speak here of "scenario". The best would be to formulate a research question like: among the changes observed during the 2020 lockdown, which ones were maintained during the 2021 restriction and which returned to the situation of 2019? At least we would have something verifiable that would begin to distinguish the work done from a simple study (with a scenario) and get closer to a future hypothesis (with a research question).

In L.182-193, the authors provide an essential explanation on the framework of 2019 which acts as a reference of normality and that of 2020 as a state modified by a sudden change, the situation of 2021 being then compared to these two states. The explanations provided seem correct, however the L.189-190 assimilate a behavior to a numerical and objectivable data such as weight and height. This seems excessive to me when we know that they are going to ask respondents for intimate details that are not note-taken. I suggest here a little more caution and a little less assertiveness.

In L.369, the title "cooking skills" suggests that we will study these cooking skills as objects of knowledge. In reality, it is a question of collecting not the skills of the respondents but the representation they have of them. The title should therefore be changed to "Self-perception of cooking skills". In this paragraph, the authors then speak alternately of "self-assessment" and "self-perception of improvement". So they report on a dynamic and not a state of skills from the point of view of the respondents, and this point should be better highlighted.

In L.417, the authors introduce the notion of "future educational action". An idea should be given of the content and form that such actions would take. In addition, it should be noted that some of the changes made during the 2020 lockdown were unstable. While some seem to spontaneously maintain and stabilize, others tend to gradually return to normal. And educational actions will have to take this dichotomy into account. On what is maintained spontaneously, we can just remind the interest of maintaining them without the need to act strongly. At the contrary, on the others, without targeted and determined action, the risk of a return to normality in a term to be determined seems high. The article highlights these two categories of change and I suggest that the authors revise their discussion to make these findings clearer. This point is a major modification of the article: it makes it possible to produce a hypothesis that will become testable in future work. In other words, the research question, by the answers obtained, produces a hypothesis for the future.

In L.446-447, the authors mention future work as I suggested in my previous revision. However, the question should be asked of when to conduct this future research, in relation to the processes concerned, assuming the absence of new COVID restrictions. Indeed, until now, the timing of data collection has been dictated by public management measures for coping the pandemic. Future data collection should be positioned in a logical duration to apprehend the processes studied, outside the influence of the pandemic.

In L.447, in the same logic, talking about "motivations" seems insufficient. It should be observed whether some behaviours return to normal while others remain close to the changes of the lockdown period. And thus test the hypothesis that the present work makes it possible to formulate.

In L.459, at the end of this discussion, as this perspective of targeted educational action is repeated, the need for precision (mentioned in my previous comment) is confirmed. As well as the need to act on some of these changes if we evaluate them positively and not be satisfied with a spontaneous evolution that would make them fade more or less quickly. Moreover, a reference should be made to actions for correcting inappropriate changes.

In L.465, I strongly disagree with this conclusion. In my opinion, the conclusion should return to the situation of 2021 which shows a very interesting in-between: in 2021 the results combine confirmations of 2020 and returns to normal of 2019. As a result, if desirable behaviours are to be maintained, targeted educational actions will be needed and not just rely on spontaneous changes. The article makes it possible to produce the hypothesis to be worked on in future work: this is its main merit and it deserves to be clearly indicated.

Corrections to be made:

In my opinion, L.215-217 should be deleted since the desired clarifications on the study site were made in L.93-97. In addition, they have nothing to do in a 2.4 on data analysis.

Reviewer 3 Report

The reply to my comments is not convincing. First of all, concerning sample structure, the Authors write:

"We understand that this appears with relative frequency as you point out. However, we do not feel that this is necessary, given that, the data are weighted so as to be adjusted to the real population in accordance with the variables of sex, age and level of education."

It is necessary in my opinion, because the Authors compare two samples. If the structure of the samples is comparable, the comparison is fine. But if it is not, the deviations might also be influenced by the differences in income, age, education, etc. It is not about transferability of the results, this is not possible (convenience sample).

There is still room for improvement in view of explaining the empirical design of the studies. "Indirect observations" is not the only issue I mentioned, the question is, how did you develop the study design in the first place (if no pre-studies were available)? You mentioned that you used sources Acosta-Banda et al. [41] or Napitupulu [42], but these papers are not explaining how you developed your study design and questionnaire. It is not surprising that you discussed it with your team members, but where are the scales coming from? Did you rely on brainstorming only?

In particular due to this shortcoming I really want to emphasize the necessity of presenting the questionnaire to the scientific community.

I disagree that the readability of the Tables should be left as it is. Authors' reply is really not sufficient. What is the problem to mark significant results with *,**,*** as this is common practice and makes everything much easier to be understood? It is not about pleasing my opinion, it is about improving readability.

Authors' answer to my comment on Fig. 1 was: "In this second phase of the questionnaire, following the recommendations of our colleagues in the nutrition and gastronomy fields and following the agreement with the work group, it was decided to eliminate this question. " In this case, they should drop the results at all. Why should one include results from a study referring to the first lockdown only. The core of the paper is the change in behavior.

The reply concerning season doesn't convince me at all. It is really necessary to discuss this limitation because if the season is responsible for the change in behavior, the comparison is (still) meaningless. This issue was not addressed.

I don't understand the reply to my comment "The Authors should really critically re-analyze their results (not only these one) and thoroughly rethink if the results might be due to a real change in behavior or due to other limitations." So what changes did the Authors make, did they really rethink the results and interpretations? This is not clear to me, I don't want to go through all modifications in the text to evaluate if this issue was considered it or not.

Dear Editors,

the adapted version of the manuscript is not satisfiying at all, im my opinion. I don't think that the issues I mentioned in my first are un-addressable. So I really don't understand why they refrained from, e.g., including the structure of the sample. Therefore, I suggest major revision although I tended towards rejecting the paper.

Kind regards

Oliver Meixner
